# When GNNs Fail: Quantifying and Overcoming Temporal Correlation Volatility in Time Series

**Chen Shao** [1] **Zhenyi Zhu** [2] **Tobias Käfer** [1]

## Abstract

Graph Neural Networks (GNNs) for multivariate time series forecasting model series as nodes and pairwise temporal correlations as edges under a static topology assumption. We study GNN representational power under both static and dynamic settings and identify critical limitations. We introduce Temporal Correlation Volatility (TCV), a model-agnostic metric for the distributional evolution of latent graph structures, and link it to performance degradation: popular models, including Transformers, generalize poorly in high-TCV regimes and are often beaten by structure-agnostic baselines. We propose Graph Layer for Inference in Dynamic Environments (GLIDE), a GNN layer with two mechanisms: (D1) Path-based Message Passing, capturing path-based neighborhoods, and (D2) Static and Dynamic Propagation Separation, identifying optimal dynamics via local static approximation. On synthetic and real-world benchmarks, GLIDE improves performance by up to $45.6\%$, with the largest gain reaching $85.7\%$. Code: https://github.com/ChenS676/GLIDE.

## 1. Introduction

Multivariate time series forecasting (MTSF) is a key challenge in domains such as energy management, weather prediction, and financial modeling (12; 10). Recently, Graph Neural Networks (GNNs) have advanced this field by modeling variables as nodes in a relational graph (18). In this framework, learning the graph topology, including connectivity and relation strength is crucial, as it governs spatial message passing and information aggregation across variables. An inaccurate or static topology can introduce spuri-

*Table 1.* MAE degradation ($\Delta\text{MAE} = \text{MAE}_{\text{d}} - \text{MAE}_{\text{s}}$) under static (Syn-S) and dynamic (Syn-D) conditions. MAE: lower is better.

| Family | Method | Syn-S | Syn-D | $\Delta$MAE (%) |
|---|---|---|---|---|
| | TCV | 0.296 | 0.874 | |
| SpatialGNN | MTGNN (18) | 0.427 | 0.508 | $-19.0\% \downarrow$ |
| | TPGNN (7) | 0.456 | 0.578 | $-26.8\% \downarrow$ |
| SpectralGNN | StemGNN (2) | 0.497 | 0.576 | $-15.9\% \downarrow$ |
| | FourGNN (21) | 0.651 | 0.828 | $-27.2\% \downarrow$ |
| Transformer | Autoformer (16) | 0.650 | 0.888 | $-36.6\% \downarrow$ |
| | Reformer (13) | 0.647 | 0.855 | $-32.1\% \downarrow$ |
| Structure-free | MLP | 0.809 | 0.927 | $-14.6\% \downarrow$ |

ous correlations, ultimately weakening the model's ability to capture complex dependencies.

Researchers typically construct graphs using predefined adjacency matrices or infer latent topology from global similarity measures such as cosine similarity, Gaussian kernels (6), or Pearson correlation (2), averaged over the full historical sequence (11; 7). This implicitly assumes a static topology and has become standard in many modern architectures (18; 7; 21). However, real-world dependencies often evolve over time, making global estimation a poor approximation of inherently dynamic structures.

This limitation becomes pronounced in highly dynamic environments where inter-variable relationships undergo abrupt structural shifts (12). For example, power grid dependencies may change due to failures or demand fluctuations, while financial correlations can shift after macroeconomic shocks (10). Existing methods fail to explicitly model such temporal variability, leading to substantial forecasting degradation. As shown in Tab. 1, performance drops can reach up to 36.6% for popular Transformer-based models (16).

To address these limitations, we study graph forecasting under dynamic topology and propose GLIDE, a GNN layer for evolving dependencies. Our contributions are:

- **Quantifying Topology Dynamics.** We introduce *Temporal Correlation Volatility* (TCV), a metric for measuring topology shifts in multivariate time series and relating them to forecasting degradation.

- **The GLIDE Architecture.** We propose GLIDE, a novel GNN layer that models higher-order interactions and decomposes persistent and transient graph dynamics to improve expressivity under evolving conditions.

- **Extensive Evaluation.** Experiments on eight benchmarks against extensive baselines show that GLIDE improves

---

[1]Karlsruhe Institute of Technology, Germany [2]The Hong Kong University of Science and Technology, Hong Kong SAR. Correspondence to: Chen Shao <cc7738@kit.edu>.

*Proceedings of the $2^{nd}$ ICML Workshop on Foundation Models for Structured Data*, Seoul, South Korea. 2026. Copyright 2026 by the author(s).

forecasting performance by up to 85.7% in highly dynamic settings while remaining competitive on static datasets.

## 2. Preliminaries

In this section, we present problem formulation and key definitions that we use in our work. Let $\mathbf{x}_t \in \mathbb{R}^N$ be a multivariate observation at time $t$, with $x_t[i]$ as the $i$-th variable. We define the look-back window of length $T$ as $\mathbf{X} = [\mathbf{x}_{t-T+1}, \ldots, \mathbf{x}_t] \in \mathbb{R}^{N \times T}$. We aim to learn a mapping $f : \mathbf{X} \to \mathbf{Y}$, where target $\mathbf{Y}$ is either a single-step vector $\mathbf{x}_{t+1}$ or multi-horizon sequence $\{\mathbf{x}_{t+1}, \ldots, \mathbf{x}_{t+P}\}$. From a graph-based perspective, $\mathbf{X}$ is a temporal graph sequence, where variables are represented as nodes and cross-variable dependencies are encoded in an adjacency matrix. Formally, the graph series is $\mathcal{G}(t) = (\mathcal{V}, \mathcal{E}_t, \mathbf{X}_t, \mathbf{A}_t)$, where $\mathcal{V}$ is the vertex set, $\mathbf{X}_t \in \mathbb{R}^{N \times B}$ are node features from a sliding window of length $B$ and $\mathbf{A}_t \in \mathbb{R}^{N \times N}$ is the sparsified weighted adjacency structure.

**Definition 2.1** (Topology-based Graph). The adjacency matrix $\mathbf{A} \in \mathbb{R}^{N \times N}$ is constructed as a $k$-nearest neighbor ($k$-NN) graph, where each entry $\mathbf{A}_{ij}$ is defined as $a_{ij}$ if $v_j \in \mathcal{N}_k(v_i)$ and 0 otherwise. $\mathcal{N}_k(v_i)$ denotes the set of $k$ nodes most similar to $v_i$ according to a chosen distance metric.

In the absence of a predefined topology, the adjacency structure is inferred from the empirical properties of the observed signals.

**Definition 2.2** (Similarity-based Graph). Given a data matrix $\mathbf{X} \in \mathbb{R}^{N \times T}$, let $\tilde{\mathbf{X}}$ denote row-wise standardized node signals. The similarity-based adjacency matrix $\hat{\mathbf{A}} \in \mathbb{R}^{N \times N}$ is defined via empirical correlation: $\hat{a}_{ij} = \frac{1}{T} \sum_{t=1}^{T} \tilde{x}_{it} \tilde{x}_{jt}$, implying $\hat{\mathbf{A}} = \frac{1}{T} \tilde{\mathbf{X}} \tilde{\mathbf{X}}^\top$ Here, $\hat{a}_{ij} \in [-1, 1]$ is the Pearson correlation coefficient, measuring the linear dependence between node signals $v_i$ and $v_j$.

### 2.1. Temporal Correlation Volatility

**Definition 2.3** (Temporal Correlation Volatility). Given a sliding-window similarity sequence $\{\hat{\mathbf{A}}_t\}_{t=1}^{T}$ with $\hat{\mathbf{A}}_t \in [-1, 1]^{N \times N}$ (Def. 2.2), the TCV is the average rate of topological change in Frobenius norm $\|\cdot\|_F$, where each $\hat{\mathbf{A}}_t$ is normalized per variable, ensuring $0 \leq \text{TCV} < 1$.

$$\text{TCV} = \frac{1}{2(T-1)N} \sum_{t=2}^{T} \|\hat{\mathbf{A}}_t - \hat{\mathbf{A}}_{t-1}\|_F, \quad (1)$$

**TCV in Practice.** The TCV distinguishes between structurally static regimes (TCV $\to$ 0) and high-volatility regimes (TCV $\to$ 1). For example, the *Electricity* dataset exhibits TCV $\approx$ 0.194, while *Solar* and *German Energy* datasets attain TCV $>$ 0.5, capturing major disruptions such as the Crimea crisis. By construction, TCV measures linear correlation drift; non-linear dependencies, lagged relationships, and causal shifts fall outside its scope.

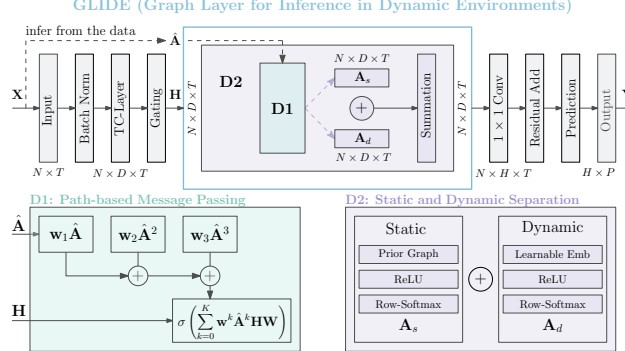

*Figure 1.* Framework of GLIDE. It consists of $K$ TC-layers, 1 GLIDE layer and a prediction layer. Inputs are first transformed by normalization, then passed to TC-Layer followed by the GLIDE layer. Each layer has residual connections and is skip-connected to the output.

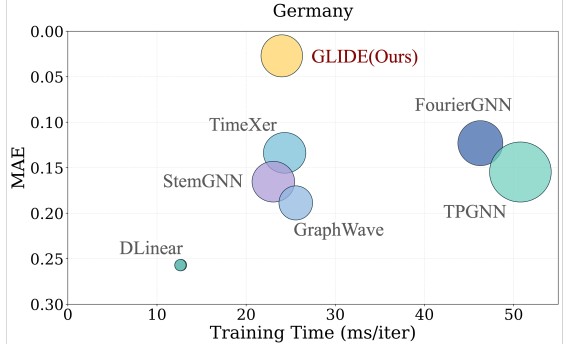

*Figure 2.* Model complexity comparison. Bubble areas represent GPU memory usage. Full complexity analysis in App. D.

### 2.2. Connecting Topology Dynamics and Performance

We investigate performance discrepancies across four dominant architecture categories under static and dynamic structural regimes. As shown in Tab. 1, spatial, spectral and Transformer architectures remain competitive in static regimes but suffer significant degradation on dynamic datasets. Transformer models experience the most acute collapse (up to 36.6%), suggesting that implicit graph induction via self-attention is highly susceptible to structural instability. This motivating example reveals two critical limitations: (i) existing architectures fail to maintain structural robustness on high-TCV datasets; and (ii) robustness is intrinsically tied to the model's *dynamic expressivity*.

## 3. GLIDE: Graph Layer for Inference in Dynamic Environments

Motivated by these limitations, we develop a novel layer centered on two architectural designs: (D1) Path-based Message Passing and (D2) Static and Dynamic Propagation Separation. We introduce and justify this layer theoretically and then introduce our full model in Sec. 3.3.

### 3.1. (D1) Path-based Message Passing

**Intuition.** When cross-variable correlations exhibit high volatility, direct edges in $\hat{\mathbf{A}}$ become susceptible to stochas-

tic oscillations and transient noise. In contrast, indirect dependencies — particularly those derived from higher-order topological neighborhoods — often emerge as more stable indicators of underlying dynamics. For instance, during the 2021–2022 European energy crisis, coal and various renewable energy sources maintained a systematic coupling mediated through the fossil fuel market (e.g., Coal→Carbon→Renewables) rather than through direct correlations. In GLIDE, each node feature $\mathbf{H}_i$ is aggregated over a path-based reachability on $\hat{\mathbf{A}}$: the $k$-hop influence of $j$ on $i$ is $[\hat{\mathbf{A}}^k]_{ij}$, and we form a $K$-th order weighted composite $R_{ij}^{(\mathrm{w})} = \sum_{k=0}^{K} w_k [\hat{\mathbf{A}}^k]_{ij}$. The local update rule is

$$\mathbf{H}_i = \sigma\left(\sum_j R_{ij}^{(\mathrm{w})} \mathbf{H}_j \mathbf{W}\right), \quad R_{ij}^{(\mathrm{w})} = \left[\sum_{k=0}^{K} w_k \hat{\mathbf{A}}^k\right]_{ij},$$

bypassing transient first-order noise to learn stable, latent connectivity.

### 3.2. (D2) Static and Dynamic Propagation Separation

While real-world similarity graphs are typically assumed static, this assumption is often violated in dynamic environments. We derive a basic form for the underlying graph structure in a dynamic setting.

**Theorem 3.1.** *Let $\{\hat{\mathbf{A}}_t\}_{t=0}^{T}$ be a sequence of similarity graphs evolving as $\hat{\mathbf{A}}_t = \hat{\mathbf{A}}_{t-1} + \mathbf{Z}_t$, where $\mathbf{Z}_t \sim \mathcal{N}(0, \mathbf{\Sigma}(t))$ is a zero-mean Gaussian perturbation with smoothly varying covariance. Then the time-varying topology at time $t$ is consistently estimated by the kernel-weighted local average:*

$$\hat{\mathbf{A}}_t = \frac{\sum_s K(s-t) \tilde{\mathbf{X}}_s \tilde{\mathbf{X}}_s^\top}{\sum_s K(s-t)}, \quad (2)$$

*where $K(\cdot)$ is a symmetric kernel weighting temporally proximate observations more heavily. Under bandwidth $h \asymp n^{-1/3}$ and $\ell_1$ regularization, the estimator recovers the true sparsity pattern of $\mathbf{\Sigma}(t)^{-1}$ with probability $1 - o(1)$. (Full conditions and proof in App. C.)*

**Justification.** Although the graph topology evolves over time, its estimation reduces to a static problem by aggregating temporally neighboring observations via kernel weighting. The second design encodes *static graph* (stable, time-invariant) separately from *dynamic graph* (time-varying shocks) during aggregation:

$$\mathbf{H} = \sigma\left(\left(\hat{\mathbf{A}}_\mathrm{s} \mathbf{W}_\mathrm{s} + \hat{\mathbf{A}}_\mathrm{d} \mathbf{W}_\mathrm{d}\right)\mathbf{H}\right) \quad (3)$$

where $\hat{\mathbf{A}}_\mathrm{s}$ is time-invariant topology and $\hat{\mathbf{A}}_\mathrm{d}$ encodes time-dependent variance. We infer $\hat{\mathbf{A}}_\mathrm{d}^{(t)}$ from temporal gradients $\nabla \tilde{\mathbf{x}}_k = \tilde{\mathbf{x}}_k - \tilde{\mathbf{x}}_{k-1}$ over a window of length $B$ (raw-signal variant in App. I):

$$\hat{\mathbf{A}}_\mathrm{d}^{(t)} = \mathrm{ReLU}\left(\frac{1}{B}\sum_{k=t-B/2}^{t+B/2}(\nabla \tilde{\mathbf{x}}_k)(\nabla \tilde{\mathbf{x}}_k)^\top \mathbf{M}_\mathrm{d}\right). \quad (4)$$

where $\mathbf{M}_\mathrm{d} \in \mathbb{R}^{N \times N}$ is a learnable mixing matrix and the ReLU echoes the non-negativity/sparsity that Graph-LASSO enforces in the theorem, capturing shock synchronization across variables.

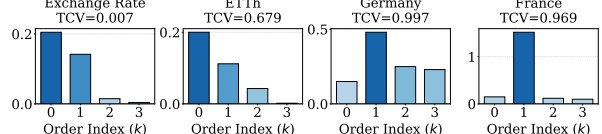

*Figure 3.* Distribution of learned coefficient magnitudes $\|w_k\|_2$ in D1. Static data (Exchange) shows decay toward lower-order terms; dynamic datasets (Germany, France) show a marked shift toward higher-order dependencies.

### 3.3. Putting Everything Together: GLIDE

GLIDE consists of three key components: **(C1)** a Temporal Convolution Layer, **(C2)** a GLIDE Layer and **(C3)** a Prediction Layer. See Fig. 1 for the full pipeline.

**C1: Temporal Convolution Layer (TC-Layer).** The TC-Layer captures a node's piecewise-smooth temporal dynamics using dilated causal convolution networks (22). Given a 1D input sequence $\mathbf{x} \in \mathbb{R}^T$ and filter $f \in \mathbb{R}^K$, the dilated causal convolution at step $t$ is:

$$(x *_d f)(t) = \sum_{s=0}^{K-1} f(s) x(t - d \cdot s), \quad (5)$$

where $d$ is the dilation factor. Stacking layers with exponentially increasing dilation yields an exponentially growing receptive field.

**C2: GLIDE Layer.** The aggregation layer updates node embeddings using an adjacency decomposed into two complementary parts: (i) static $\hat{\mathbf{A}}_\mathrm{s}$ for long-term topology and (ii) dynamic $\hat{\mathbf{A}}_\mathrm{d}$ for transient dependencies:

$$\mathbf{H} = \sigma\Big(\underbrace{\sum_{k=0}^{K} \mathbf{w}_\mathrm{s}^k \mathbf{A}_\mathrm{s}^k \mathbf{W}_\mathrm{s}}_{\text{static}}\Big)\mathbf{H} + \Big(\underbrace{\sum_{k=0}^{K} \mathbf{w}_\mathrm{d}^k \mathbf{A}_\mathrm{d}^k \mathbf{W}_\mathrm{d}}_{\text{dynamic}}\Big)\mathbf{H}.$$

(6)

**C3: Prediction Layer.** The final node representations $\mathbf{H}^{(\mathrm{f})} \in \mathbb{R}^{N \times F}$ from the GLIDE layer are mapped to future outputs via a 1D convolutional projection $\hat{\mathbf{Y}} = \mathbf{H}^{(\mathrm{f})}\mathbf{W}_c$, where $\mathbf{W}_c \in \mathbb{R}^{F \times P}$ is a learnable weight. Here, $\hat{\mathbf{Y}} \in \mathbb{R}^{N \times P}$ represents the predicted sequences for all nodes.

## 4. Empirical Evaluation

We address the following research questions: **(RQ)** On real-world time-series data with difficult dynamic topology, to what extent does GLIDE improve forecasting?

**Datasets.** We consider two TCV regimes on real-world benchmarks: a static and a dynamic configuration. For the static setting, we use the benchmarks *Electricity* and *Solar*(7). To further assess performance under high dynamics, we include two recent energy datasets, *Germany* and *France* from (12). All datasets use a 70/20/10 split. The synthetic data setup for Tab. 1 is detailed in App. H.1.

*Table 2.* REAL-STATIC-DYNAMIC: Mean $\pm$ std MAE/RMSE ($h = 12$). The top three results are highlighted. All std dev $< 0.009$. More details are in Tab. 5

| Method | Electricity TCV = 0.194 | | Solar TCV = 0.323 | | Germany TCV = 0.997 | | France TCV = 0.969 | | Rank |
|---|---|---|---|---|---|---|---|---|---|
| | MAE | RMSE | MAE | RMSE | MAE | RMSE | MAE | RMSE | |
| TPGNN | 0.055 | 0.080 | 0.123 | 0.214 | 0.099 | 0.173 | 0.089 | 0.158 | 3 |
| GWaveNet | 0.094 | 0.140 | 0.183 | 0.238 | 0.013 | 0.028 | 0.012 | 0.025 | 6 |
| MTGNN | 0.077 | 0.113 | 0.151 | 0.207 | 0.016 | 0.034 | 0.012 | 0.023 | 4 |
| FourierGNN | 0.051 | 0.077 | 0.120 | 0.162 | 0.110 | 0.186 | 0.096 | 0.164 | 2 |
| StemGNN | 0.070 | 0.101 | 0.176 | 0.222 | 0.179 | 0.285 | 0.148 | 0.206 | 5 |
| TimeMixer | 0.091 | 0.147 | 0.166 | 0.211 | 0.181 | 0.314 | 0.167 | 0.279 | 13 |
| PatchTST | 0.212 | 0.309 | 0.188 | 0.305 | 0.153 | 0.332 | 0.219 | 0.408 | 15 |
| Autoformer | 0.056 | 0.083 | 0.150 | 0.193 | 0.204 | 0.376 | 0.165 | 0.263 | 8 |
| FEDformer | 0.055 | 0.081 | 0.139 | 0.182 | 0.271 | 0.396 | 0.220 | 0.291 | 11 |
| Informer | 0.070 | 0.119 | 0.151 | 0.199 | 0.283 | 0.324 | 0.137 | 0.217 | 7 |
| TCN | 0.057 | 0.083 | 0.176 | 0.222 | 0.187 | 0.287 | 0.172 | 0.260 | 9 |
| LSTNet | 0.075 | 0.138 | 0.148 | 0.200 | 0.193 | 0.346 | 0.177 | 0.263 | 12 |
| DLinear | 0.058 | 0.092 | 0.257 | 0.313 | 0.266 | 0.368 | 0.196 | 0.259 | 16 |
| VAR | 0.096 | 0.155 | 0.175 | 0.222 | 0.243 | 0.381 | 0.177 | 0.260 | 17 |
| **GLIDE** | **0.010** | **0.051** | **0.024** | **0.061** | **0.005** | **0.020** | **0.010** | **0.020** | **1** |

**Baselines.** We compare against 15 representative forecasting models spanning six methodological families. *Spatial GNNs*: GWaveNet [17], MTGNN [18] and TPGNN [7]; *Spectral GNNs*: StemGNN [2] and FourGNN [21]; *Transformers*: Informer [24], Autoformer [16], FEDformer [25] and Reformer [4]; *Sequence Models*: TCN [1] and LSTNet [5]; *Decomposition-based Models*: TimeMixer [14] and PatchTST [9]; and the classical *VAR* and DLinear [23].

**Metrics.** We report MAE and RMSE at horizons $h = 3, 6, 12$ over five random seeds with grid-search hyperparameter tuning (details in App. H).

### 4.1. RQ: Evaluation under Dynamic Topologies

Tab. 2 reports MAE/RMSE across four datasets comparing against 14 diverse baselines. GLIDE consistently achieves state-of-the-art performance in every category (Rank = 1). On the Electricity dataset, GLIDE reduces MAE from 0.051 to 0.010 — an absolute reduction of 0.041 and a 80.3% relative gain over the runner-up FourGNN. Similarly, on Germany, GLIDE substantially improves over MTGNN, reducing MAE from 0.013 to 0.005 and RMSE from 0.028 to 0.020, corresponding to relative improvements of 61.54% and 28.57%, respectively. We provide additional experiments on synthetic and static real-world benchmarks in the Appendix. Here, we briefly summarize the main conclusion: even in static time series forecasting scenarios where the underlying structure does not change, GLIDE is still able to maintain its accuracy.

### 4.2. Ablation Studies

**Impact of two Designs** ($D_1, D_2$)**.** Fig. 4 compares GLIDE with two ablated variants. Figs. 4(a)-(b) show that remov-

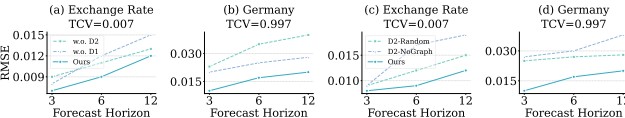

*Figure 4.* Ablation study of GLIDE. (a–b) Comparison with base ablations (w.o. $D_1$ and $D_2$). (c–d) Sensitivity analysis of $D_2$ removing dynamic topology ($D_2$-NoGraph) and a random matrix baseline ($D_2$-Random).

ing $D_1$ yields substantial performance degradation on both static and dynamic datasets. Fig. 4(b) indicates that under the high-TCV setting (Germany), $D_2$ emerges as the primary performance driver; its removal results in nearly twofold the error compared to other variants.

**Impact of Dynamic Graph Designs** ($D_2$)**.** In Figs. 4(c)-(d), our full approach achieves the best MAE/RMSE across all horizons: (1) replacing $\mathbf{A}_d$ with a random graph ($D_2$-Random) yields slightly higher errors at longer horizons; (2) removing the dynamic graph ($D_2$-NoGraph) produces the weakest results with substantially larger RMSE.

## 5. Conclusion

We characterized the representational power of GNNs for time series of static or dynamic topology. We highlighted the limitations of current methods and proposed GLIDE with theoretically grounded components. Together, our method and the linear TCV establish a solid foundation for dynamic topology modeling. Experiments show that GLIDE is robust to topology variations and outperforms strong baselines.

**Disclosure of AI Assistance.** The authors employed AI

tools for grammatical refinement and LATEX table optimization.

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

# A. Notation

*Table 3.* Summary of notation used throughout the paper.

| Symbol | Description |
|---|---|
| $N$ | Number of variables (nodes in the graph) |
| $T$ | Length of the look-back window |
| $P$ | Forecast horizon (number of future steps to predict) |
| $B$ | Length of the sliding window for node features ($B \leq T$) |
| $\mathbf{x}_t$ | Multivariate observation at time $t$, $\mathbf{x}_t \in \mathbb{R}^N$ |
| $x_t[i]$ | The $i^{\text{th}}$ variable (node feature) in $\mathbf{x}_t$ |
| $\mathbf{X}$ | Input look-back sequence, $\mathbf{X} \in \mathbb{R}^{N \times T}$ |
| $\mathbf{Y}$ | Target future value(s) |
| $\mathcal{G}(t)$ | Temporal graph at time $t$ |
| $\mathcal{V}$ | Set of vertices (variables), $|\mathcal{V}| = N$ |
| $\mathcal{E}_t$ | Set of edges at time $t$ |
| $\mathbf{A}_t$ | Sparsified weighted adjacency matrix |
| $\hat{\mathbf{A}}$ | Similarity-based adjacency matrix, $\hat{\mathbf{A}} = \frac{1}{T}\tilde{\mathbf{X}}\tilde{\mathbf{X}}^\top$ |
| $\hat{a}_{ij}$ | Pearson correlation coefficient between nodes $v_i$ and $v_j$ |
| $f(\cdot)$ | Prediction function mapping $\mathbf{X}$ (or $\mathcal{G}(t)$) to $\mathbf{Y}$ |

# B. Related Work

**GNN in Time Series.** SpatialGNNs have shown strong performance in time series by modeling relational structures through graph topology ([18]; [20]; [12]). Li et al. ([6]) pioneered diffusion convolution to capture spatial dependencies from data, while Wu et al. ([17]) introduced adaptive adjacency learning to uncover hidden correlations. Subsequent works enhance correlation modeling via spectral kernels or polynomial graph encoders ([19]; [2]; [21]; [7]).

**Graph Structure Learning.** Existing GSL methods for time series generally fall into three categories: *latent-static* approaches (e.g., MTGNN) that learn time-invariant spatial priors via node embeddings; *stochastic* models (e.g., GTS) that use Gumbel-Softmax sampling for probabilistic topology; and *spectral operators* (e.g., FourGNN) that optimize efficiency through frequency-domain kernels. In contrast, our framework instantiates an adaptive learning framework to capture dynamic inter-variable dependencies with evolving structural shifts.

# C. Details of Theorems

**Theorem C.1.** *Let $\{\mathbf{A}_t\}$ be independent random adjacency structures with $\mathbf{A}_t \sim \mathcal{N}(0, \Sigma(t))$ associated with a time index $t = 0, \frac{1}{n}, \frac{2}{n}, \ldots, 1$. Each $\mathbf{A}_t$ corresponds to an undirected graph $\mathcal{G}(t)$. Under the assumption that the law $\mathcal{L}(\mathbf{A}_t)$ varies smoothly with $t$, the structure of $\mathcal{G}(t)$ is determined by the zero pattern of the precision matrix $\Sigma(t)^{-1}$. This framework applies to the global similarity evolution: $\hat{\mathbf{A}}_0 \sim \mathcal{N}(0, \Sigma(0))$, $\hat{\mathbf{A}}_t = \hat{\mathbf{A}}_{t-1} + \mathbf{A}_t$. In the non-i.i.d. setting, we estimate $\Sigma(t)$ at time $t$ by:*

$$\widehat{\Sigma}_n(t) = \arg\min_{\Sigma \succ 0} \left\{ \text{tr}\left(\Sigma^{-1}\widehat{S}_n(t)\right) + \log|\Sigma| + \lambda\|\Sigma^{-1}\|_1 \right\}, \tag{7}$$

*where $\widehat{S}_n(t) = \frac{\sum_s w_{st} Z_s Z_s^\top}{\sum_s w_{st}}$ is a kernel-weighted covariance matrix with $w_{st} = K\left(\frac{|s-t|}{h_n}\right)$.*

*Proof.* We establish probabilistic bounds governing the convergence of similarity-based graph estimators. Our data consists of $n$ multivariate observations $\mathbf{x}_k \in \mathbb{R}^N$ sampled at $x = 0, 1/n, \ldots, 1$, with $\mathbf{x}_k \sim \mathcal{N}(0, \Sigma_k)$. The kernel weights $\ell_k(x_0)$ are:

$$\ell_k(x_0) = \frac{2}{nh} K\left(\frac{x_k - x_0}{h}\right) \approx \frac{K\left(\frac{x_k - x_0}{h}\right)}{\sum_{k=1}^{n} K\left(\frac{x_k - x_0}{h}\right)}. \tag{8}$$

**Lemma C.2** (Global Bias Bound). *Suppose $\max_{i,j} \sup_x |\sigma''(x, i, j)| \leq C$. Then for all $x \in [0,1]$: $\max_{i,j} |\mathbb{E}[\hat{a}_{ij}(x)] - \sigma_{ij}(x)| = O(h)$.*

**Lemma C.3** (Large Deviation of Graph Structures). *For sufficiently small $\epsilon$:* $\mathbb{P}(|\hat{a}_{ij} - \mathbb{E}[\hat{a}_{ij}]| > \epsilon) \leq \exp\left\{-Cnh\epsilon^2\right\}.$

By applying Markov's inequality to the exponential of the weighted sum, choosing optimal $t = \frac{\epsilon}{4\Phi_2}$, and using the fact that $\Phi_2 \approx \frac{C_1(\sigma_i^2\sigma_j^2 + \sigma_{ij}^2)}{h}$, we obtain:

$$\mathbb{P}(\dots) \leq \exp\left(-\frac{3nh\epsilon^2}{20C_1(\sigma_i^2\sigma_j^2 + \sigma_{ij}^2)}\right).$$

Thus, the Global Similarity Graph $\hat{\mathbf{A}}$ is a persistent estimator of the underlying precision structure $\Sigma(x_0)^{-1}$ as $nh \to \infty$. $\square$

**Theorem C.4** (Dynamic Topology Identification, Full Version). *Consider a multivariate time series $\mathbf{X} \in \mathbb{R}^{N \times T}$ with an underlying graph topology evolving as $\hat{\mathbf{A}}_t = \hat{\mathbf{A}}_{t-1} + \mathbf{A}_t$, where $\mathbf{A}_t \sim \mathcal{N}(0, \boldsymbol{\Sigma}(t))$. Let $\Theta(t) = \boldsymbol{\Sigma}(t)^{-1}$. Under: (1) **Temporal Smoothness**: $\max_{i,j} \sup_t |\sigma_{ij}''(t)| \leq C$; (2) **Sparsity**: $|E_t| = s$ satisfying $(N + s) = o(n^{2/3}/\log N)$; (3) **Optimal Bandwidth**: $h \asymp n^{-1/3}$; the time-varying topology is identified via:*

$$\hat{\Theta}_t = \arg\min_{\Theta \succ 0}\left\{tr(\Theta\hat{\mathbf{A}}_t) - \log|\Theta| + \lambda_n\|\Theta\|_1\right\}, \tag{9}$$

*where $\hat{\mathbf{A}}_t = \frac{\sum_s w_{st}\mathbf{A}_s}{\sum_s w_{st}}$. Then with probability $1 - o(1)$, $\hat{\Theta}_t$ is **sparsistent**, correctly recovering the zero-pattern of $\Theta(t)$.*

## D. Time Complexity Analysis

With a given sequence $\mathbf{X} \in \mathbb{R}^{N \times T}$, let $F$ denote the hidden feature dimension, $|\mathcal{E}|$ the number of edges, $k$ the kernel size and $T_p$ the prediction horizon. The temporal convolution stage (S1) takes $O(NkT\log T)$. The GLIDE layer (S2) takes $O(k|E|F)$ for order $k$. The prediction stage (S3) requires $O(NFT_p)$. Overall, the total complexity is $O(NkT\log T + k|E|F + NFT_p)$, with memory cost dominated by $O(NF)$ for embeddings. Fig. 2 shows that GLIDE achieves better efficiency than FourierGNN and TPGNN.

## E. Extended Related Work

**Graph Topology Identification (GTI).** When the graph structure in time series is not available, topology identification is essential. In static settings, under mild assumptions (e.g., Gaussian), the problem reduces to estimating the sample covariance matrix, which can be interpreted as an adjacency matrix with self-loops (18). Structural equation models (SEM) formulate node values as linear functions of neighbors with noise, enabling adjacency estimation via regression. With prior constraints such as smoothness or sparsity, this leads to Graph LASSO formulations (3). For dynamic settings, empirical covariance is updated via exponentially-weighted moving average (8).

## F. More Details of Motivating Example

**Synthetic Data.** We generate synthetic data with various TCV values following Liu et al. (7). We construct $N_w$ graphs $\{\mathcal{G}^{(1)}, \dots, \mathcal{G}^{(N_w)}\}$, where topology complexity $T_{\mathcal{G}} \leq N_w$ controls the switching period. When $N_w = T_{\mathcal{G}}$ it yields a static topology; smaller $T_{\mathcal{G}}$ produces faster switching and higher TCV. Details are in App. H.1.

## G. Extended Empirical Evaluation

**(RQ1)** In a synthetic setting where data generation and topology dynamics are known, does GLIDE improve performance?
**(RQ2)** On commonly used datasets with (near-)static topology, does GLIDE preserve strong performance?

### G.1. RQ1: Synthetic Benchmarks

We evaluate GLIDE on synthetic data spanning low-to-high topology dynamics (TCV) and compare it against seven models from three dominant categories: *Spatial GNNs*, *Spectral GNNs* and *Transformers*. Table 4 reports the mean MAE/RMSE of three prediction horizons ($h = 3, 6, 12$). GLIDE consistently outperforms all competing baselines, attaining the lowest RMSE/MAE in all six columns. This dominance persists across various topology regimes and over short to mid horizons. As the graph topology complexity (measured by TCV) increases, all methods exhibit performance degradation; however, GLIDE maintains a significant lead. Specifically, GLIDE achieves up to a 23.25% performance improvement over the

*Table 4.* SYNTHETIC: Mean ± std MAE/RMSE (M/R) for Static(Syn-S) and Dynamic(Syn-D) categories with varying horizons.

| Methods | Me. | Syn-S (TCV=0.296) | | | Syn-D (TCV=0.874) | | | Rk |
|---|---|---|---|---|---|---|---|---|
| | | 3 | 6 | 12 | 3 | 6 | 12 | |
| Informer | M | $.423_{\pm.019}$ | $.567_{\pm.023}$ | $.672_{\pm.029}$ | $.613_{\pm.031}$ | $.772_{\pm.036}$ | $.897_{\pm.063}$ | 8 |
| | R | $.552_{\pm.020}$ | $.726_{\pm.025}$ | $.878_{\pm.026}$ | $.764_{\pm.027}$ | $.943_{\pm.034}$ | $1.045_{\pm.047}$ | |
| Autoformer | M | $.402_{\pm.015}$ | $.553_{\pm.019}$ | $.650_{\pm.023}$ | $.592_{\pm.025}$ | $.745_{\pm.029}$ | $.890_{\pm.052}$ | 7 |
| | R | $.543_{\pm.016}$ | $.736_{\pm.021}$ | $.902_{\pm.022}$ | $.742_{\pm.022}$ | $.948_{\pm.028}$ | $1.085_{\pm.041}$ | |
| Reformer | M | $.628_{\pm.028}$ | $.684_{\pm.028}$ | $.850_{\pm.036}$ | $.621_{\pm.032}$ | $.689_{\pm.027}$ | $.722_{\pm.051}$ | 6 |
| | R | $.903_{\pm.032}$ | $.958_{\pm.033}$ | $1.038_{\pm.031}$ | $.927_{\pm.033}$ | $.979_{\pm.035}$ | $.972_{\pm.044}$ | |
| FourGNN | M | $.447_{\pm.020}$ | $.582_{\pm.024}$ | $.653_{\pm.028}$ | $.609_{\pm.031}$ | $.759_{\pm.036}$ | $.948_{\pm.067}$ | 4 |
| | R | $.588_{\pm.021}$ | $.764_{\pm.026}$ | $.843_{\pm.025}$ | $.756_{\pm.027}$ | $.935_{\pm.033}$ | $1.157_{\pm.053}$ | |
| StemGNN | M | $.421_{\pm.019}$ | $.553_{\pm.023}$ | $.576_{\pm.025}$ | $.577_{\pm.029}$ | $.698_{\pm.033}$ | $.782_{\pm.055}$ | 3 |
| | R | $.561_{\pm.020}$ | $.733_{\pm.025}$ | $.807_{\pm.024}$ | $.713_{\pm.025}$ | $.884_{\pm.031}$ | $.951_{\pm.043}$ | |
| TPGNN | M | $.251_{\pm.011}$ | $.302_{\pm.013}$ | $.456_{\pm.020}$ | $.431_{\pm.022}$ | $.521_{\pm.025}$ | $.543_{\pm.038}$ | 2 |
| | R | $.426_{\pm.015}$ | $.459_{\pm.016}$ | $.589_{\pm.018}$ | $.570_{\pm.020}$ | $.580_{\pm.021}$ | $.643_{\pm.029}$ | |
| DCRNN | M | $.252_{\pm.011}$ | $.305_{\pm.013}$ | $.358_{\pm.014}$ | $.419_{\pm.016}$ | $.472_{\pm.018}$ | $.509_{\pm.022}$ | 2 |
| | R | $.434_{\pm.015}$ | $.469_{\pm.016}$ | $.500_{\pm.017}$ | $.571_{\pm.018}$ | $.583_{\pm.019}$ | $.612_{\pm.023}$ | |
| **Ours** | M | $\mathbf{.246}_{\pm.008}$ | $\mathbf{.297}_{\pm.009}$ | $\mathbf{.350}_{\pm.011}$ | $\mathbf{.415}_{\pm.012}$ | $\mathbf{.461}_{\pm.014}$ | $\mathbf{.496}_{\pm.018}$ | 1 |
| | R | $\mathbf{.423}_{\pm.010}$ | $\mathbf{.458}_{\pm.011}$ | $\mathbf{.494}_{\pm.012}$ | $\mathbf{.566}_{\pm.014}$ | $\mathbf{.572}_{\pm.015}$ | $\mathbf{.602}_{\pm.019}$ | |

*Table 5.* REAL-BENCHMARK: Mean ± std RMSE/MAE(R/M) ($h = 3, 6, 12$). The top two models for each metric are highlighted.

| Methods | Me. | Exchange Rate (TCV =0.007) | | | ETTh1 (TCV =0.679) | | | Rk |
|---|---|---|---|---|---|---|---|---|
| | | 3 | 6 | 12 | 3 | 6 | 12 | |
| StemGNN | R | $.063_{\pm.009}$ | $.189_{\pm.015}$ | $.123_{\pm.014}$ | $.496_{\pm.001}$ | $.573_{\pm.009}$ | $.660_{\pm.014}$ | 7 |
| | M | $.190_{\pm.012}$ | $.290_{\pm.016}$ | $.277_{\pm.017}$ | $.349_{\pm.001}$ | $.408_{\pm.009}$ | $.476_{\pm.011}$ | |
| TimeMixer | R | $.217_{\pm.014}$ | $.277_{\pm.018}$ | $.293_{\pm.020}$ | $.379_{\pm.000}$ | $.459_{\pm.003}$ | $.515_{\pm.003}$ | 6 |
| | M | $.631_{\pm.029}$ | $.775_{\pm.036}$ | $.875_{\pm.042}$ | $.243_{\pm.001}$ | $.290_{\pm.001}$ | $.328_{\pm.001}$ | |
| FourGNN | R | $.221_{\pm.012}$ | $.268_{\pm.015}$ | $.292_{\pm.017}$ | $.508_{\pm.004}$ | $.558_{\pm.002}$ | $.611_{\pm.007}$ | 5 |
| | M | $.016_{\pm.003}$ | $.033_{\pm.005}$ | $.049_{\pm.006}$ | $.351_{\pm.004}$ | $.385_{\pm.002}$ | $.425_{\pm.007}$ | |
| MTGNN | R | $.023_{\pm.003}$ | $.045_{\pm.004}$ | $.090_{\pm.006}$ | $.457_{\pm.001}$ | $.548_{\pm.009}$ | $.640_{\pm.012}$ | 4 |
| | M | $.044_{\pm.004}$ | $.087_{\pm.006}$ | $.131_{\pm.009}$ | $.310_{\pm.002}$ | $.372_{\pm.007}$ | $.440_{\pm.003}$ | |
| TPGNN | R | $.009_{\pm.002}$ | $.018_{\pm.003}$ | $.035_{\pm.004}$ | $.445_{\pm.006}$ | $.540_{\pm.019}$ | $.627_{\pm.012}$ | 3 |
| | M | $.020_{\pm.003}$ | $.039_{\pm.004}$ | $.078_{\pm.006}$ | $.277_{\pm.004}$ | $.345_{\pm.014}$ | $.410_{\pm.008}$ | |
| GWaveNet | R | $.013_{\pm.003}$ | $.033_{\pm.004}$ | $.034_{\pm.005}$ | $.264_{\pm.004}$ | $.319_{\pm.001}$ | $.369_{\pm.003}$ | 2 |
| | M | $.078_{\pm.005}$ | $.139_{\pm.009}$ | $.124_{\pm.010}$ | $.133_{\pm.002}$ | $.164_{\pm.002}$ | $.193_{\pm.002}$ | |
| **Ours** | R | $\mathbf{.007}_{\pm.001}$ | $\mathbf{.009}_{\pm.001}$ | $\mathbf{.012}_{\pm.002}$ | $\mathbf{.171}_{\pm.009}$ | $\mathbf{.184}_{\pm.009}$ | $\mathbf{.184}_{\pm.010}$ | 1 |
| | M | $\mathbf{.005}_{\pm.001}$ | $\mathbf{.006}_{\pm.001}$ | $\mathbf{.007}_{\pm.001}$ | $\mathbf{.135}_{\pm.007}$ | $\mathbf{.135}_{\pm.008}$ | $\mathbf{.154}_{\pm.009}$ | |

strongest baselines (TPGNN, h=12), with the most pronounced gains observed in MAE. Overall, the proposed method shows improvements across various degrees of topology dynamics for short-term prediction tasks.

### G.2. RQ2: Evaluation on Common Benchmarks

Table 5 shows that GLIDE consistently outperforms six strong baselines across both benchmarks and most forecasting horizons. It achieves the best overall ranking and attains the lowest RMSE in all evaluated settings. For MAE, GLIDE yields the best performance in all but one case (ETTh1 at $h = 3$), where the difference to the strongest baseline is marginal. Compared with *GWaveNet*, the strongest competing method on ETTh1, GLIDE reduces RMSE by up to 50.1% and MAE by up to 20.2%. On the Exchange Rate benchmark, the improvements are more pronounced: GLIDE achieves up to 64.7% RMSE reduction and up to 85.7% MAE reduction over the second-best model at each horizon. Averaged across the three horizons, GLIDE reduces RMSE by 45.6% and MAE by 78.8%, confirming that the improvement is broadly sustained rather than driven by a single horizon. Across both Exchange Rate and ETTh1, mean RMSE reductions remain comparable (45.6% vs 42.5%) despite a 100× difference in TCV, indicating that the gains generalize across volatility regimes.

# H. Empirical Setup

## H.1. Evaluation on Synthetic Benchmark

### H.1.1. SYNTHETIC & REAL DATA.

Following the synthetic data generation procedure described in Sec. 2.2, we consider two topology-change regimes: a dynamic configuration (TCV = 0.874) and a static configuration (TCV = 0.296). We evaluate on widely used benchmarks, *Electricity* and *Solar* as in (7). To further assess performance under high dynamics, we also include two recent energy datasets from (12), *Germany* and *France*. We also compare our method on two commonly used datasets, *ETTh* and *Exchange Rate* (7).

### H.1.2. PREPROCESSING.

We follow the preprocessing convention of (7; 21): each variable is standardised per-channel using statistics computed on the training split only, and MAE/RMSE are reported in normalised space. Splits are 70/20/10 for the real-world benchmarks (Electricity, Solar, Germany, France, ETTh1, Exchange Rate) and 60/20/20 for the synthetic configurations. Full hyperparameter grids and dataset statistics are in App. H.

### H.1.3. DETAILS OF SYNTHETIC DATA GENERATION

At time step $t$, the signal $X(t) \in \mathbb{R}^{N \times 1}$ is sampled as $X(t) \sim \mathcal{N}\big(A(t-1)X(t-1), \sigma\big)$, where $\mathcal{N}$ denotes the normal distribution, $\sigma \in \mathbb{R}$ specifies the variance, and $X(0)$ is drawn from a discrete uniform distribution, $A(t-1)$ denotes a transition matrix indicates the underlying graph topology behind time series. We construct $N_w$ transition graphs $(\mathbf{A}_1, \ldots, \mathbf{A}_{N_w})$, each encoding the structural and transitional statistics of the series within a specific interval. Random walk sequences are generated on these graphs, forming the temporal dynamics: variables correspond to independent walks but share the same transition graph within an interval. Given a cycle length $T_{\mathcal{G}} \geq N_w$, we divide it into $N_w$ equal subintervals of length $T_s = T_{\mathcal{G}}/N_w$. Each subinterval is assigned a transition matrix, $\mathbf{A}(t) = G\Big(\Big\lfloor \frac{t \bmod T_{\mathcal{G}}}{T_s} \Big\rfloor\Big)$, with $\lfloor \cdot \rfloor$ denoting the floor function. Within each $T_s$-interval, we simulate $M$ independent random walks as the $M$-dimensional time series. The static limit corresponds to $N_w = 1$ (a single fixed transition graph), while smaller $T_s$ (equivalently, larger $N_w$ at fixed $T_{\mathcal{G}}$) induces faster switching and higher-order correlations. To further increase stochasticity and realism, we incorporate random rewiring and Bernoulli noise. We highlight two extreme regimes of this construction and generate six multivariate time series datasets of length 2400. Each dataset is split into training, validation, and test subsets with a 7:1:2 ratio. Full algorithmic details, we provide algorithm table in Algorithm 1.

---

**Algorithm 1** Generating MTS Data with the NPR Model

---

**Require:** Total length $T$, number of variables $N$, number of constant matrices $N_w$, cycle length $T_p$, standard deviation $\sigma$, matrix sparsity threshold $\delta$
**Ensure:** Synthetic MTS data $\mathbf{X} \in \mathbb{R}^{T \times N}$
 1: Generate random orthogonal matrix $\mathbf{P} \in \mathbb{R}^{N \times N}$
 2: **for** $i = 1$ to $N_w$ **do**
 3:     $\mathbf{\Sigma}_i = \mathrm{diag}(|\mathcal{N}(0,1)|, \ldots, |\mathcal{N}(0,1)|)$
 4:     $\mathbf{G}_i = \mathbf{P}^{\top} \mathbf{\Sigma}_i \mathbf{P}, \alpha = 0$
 5:     **while** sparsity$(\mathbf{G}_i) > \delta$ **do**
 6:         $\alpha \leftarrow \alpha + 0.02$; $\mathbf{G}_i[\mathbf{G}_i < \alpha] \leftarrow 0$
 7:     **end while**
 8:     $\mathbf{G}_i \leftarrow$ symmetric normalized Laplacian of $\mathbf{G}_i$
 9: **end for**
10: Initialize $\mathbf{X} \leftarrow \mathbf{0} \in \mathbb{R}^{T \times N}, T_l \leftarrow T/T_p$
11: **for** $t = 1$ to $T$ **do**
12:     **if** $(t-1) \bmod T_p = 0$ **then**
13:         Initialize $\mathbf{x} \in \mathbb{R}^N$ randomly from $\{-1, -0.5, 0.5, 1\}$
14:     **else**
15:         $\mathbf{x} \sim \mathcal{N}\big(\mathbf{G}_{(t-1) \bmod T_p/T_l} \cdot \mathbf{X}[t-1], \sigma\big)$
16:     **end if**
17:     $\mathbf{X}[t] \leftarrow \mathbf{x}$
18: **end for**

---

### H.1.4. IMPLEMENTATION

All models are trained under a standardized setup with temporal splits of 60%/20%/20%, and early stopping of epoch 5 is applied based on validation Mean Absolute Error (MAE). We adopt the same hyperparameter grids for parameters embedding sizes $\{32, 128, 512\}$, hidden sizes $\{16, 64, 128\}$, learning rates $\{10^{-3}, 10^{-4}, 10^{-5}\}$, and batch sizes $\{32, 64, 128, 512\}$.

### H.2. More Details of Tab. 2

#### H.2.1. HYPERPARAMETER TUNING.

We performed a grid search over a wide range of hyperparameter values for all reported models. Due to limited space, we only report a subset of the tuned hyperparameters, which include but are not limited to: learning rate $10^{(-2 \sim -4)}$, hidden embedding size $2^{8 \sim 10}$, batch size $2^{4 \sim 9}$, number of encoder and decoder layers $1 \sim 3$, number of attention heads $2 \sim 8$, kernel sizes $3 \sim 7$ and rolling window sizes $12 \times (1 \sim 3)$. For the baseline result, we report the best results obtained from (21; 15; 18).

### H.3. More Details of RQ2

#### H.3.1. DATASET & METRICS.

We finally evaluate GLIDE on a real-world high-TCV data set. Specifically, we select a subset from energy datasets in (12), *Germany*, *France* and two public datasets (electricity and solar). It records electricity production in Germany and France in all generation types for all available years. We use the consistent data splits (70%/20%/10%) for all models and report *Mean Absolute Error (MAE)* and *Root Mean Squared Error (RMSE)* of forecasting horizons with 12 time steps on five random seeds with enabled early stop. The datasets span a wide correlation spectrum. For instance, Germany and France exhibit larger magnitudes of TCV (near 1) compared to the classic datasets (around 0.5), indicating stronger temporal fluctuations and heterogeneous regional dynamics. For other results on dynamic datasets please refer to Tab. 7.

#### H.3.2. HYPERPARAMETER TUNING.

We performed a grid search over a wide range of hyperparameter values for all reported models. We report the tuned hyper-parameters, which include but are not limited to: learning rate $10^{(-2 \sim -4)}$, hidden embedding size $2^{8 \sim 10}$, batch size $2^{4 \sim 9}$, number of encoder and decoder layers $1 \sim 3$, number of attention heads $2 \sim 8$, kernel sizes $3 \sim 7$ and rolling window sizes $12 \times (1 \sim 3)$. For the electricity, solar and ETT datasets, we report the best results obtained from (21; 15).

*Table 6.* Dataset statistics and corresponding TCV values.

| Dataset | $N$ | Rows | Windows | $B$ | Step | TCV |
|---|---|---|---|---|---|---|
| Germany | 16 | 333,060 | 333,037 | 24 | 1 | 0.997 |
| France | 10 | 83,265 | 83,242 | 24 | 1 | 0.969 |
| Exchange Rate | 8 | 7,588 | 7,565 | 24 | 1 | 0.007 |
| Electricity | 321 | 26,304 | 26,281 | 24 | 1 | 0.194 |
| ETTh1 | 7 | 17,420 | 17,397 | 24 | 1 | 0.679 |
| Solar | 137 | 52,560 | 12,342 | 24 | 1 | 0.323 |

## I. Ablation Study

**Ablation on Dynamic Graph Design** We evaluate the incremental contributions of our two design choices under the same training protocol. Recall that **D1** removes the second (adaptive) graph and keeps a single fixed/base graph; **D2** denotes dual-graph variants with targeted edits: (*i*) **D2-Random** replaces the base graph with a random topology while retaining the adaptive branch, and (*ii*) **D2-noGraph** disables the adaptive branch while keeping the base graph. Unless otherwise noted, depth and optimization are identical across variants. For tables reporting $H \in 3, 6, 12$, the value at $H=K$ is the *prefix-mean* over the first $K$ horizons.

*Table 7.* Mean MAE/RMSE for two dynamic-correlation energy datasets. We report MAE, RMSE and MAPE at horizons 6 and 12. Best, second-best and third-best results are highlighted.

| Methods | Germany (TCV = 0.997) | | | | | | France (TCV = 0.969) | | | | | | Rank |
| | H=6 | | | H=12 | | | H=6 | | | H=12 | | | |
| | MAE | RMSE | MAPE | MAE | RMSE | MAPE | MAE | RMSE | MAPE | MAE | RMSE | MAPE | |
|---|---|---|---|---|---|---|---|---|---|---|---|---|---|
| VAR | 0.210 | 0.340 | 0.191 | 0.243 | 0.381 | 0.215 | 0.155 | 0.241 | 0.132 | 0.177 | 0.260 | 0.151 | 11 |
| DLinear | 0.230 | 0.320 | 0.201 | 0.266 | 0.368 | 0.231 | 0.170 | 0.230 | 0.140 | 0.196 | 0.259 | 0.158 | 12 |
| LSTNet | 0.165 | 0.290 | 0.145 | 0.193 | 0.346 | 0.172 | 0.150 | 0.220 | 0.129 | 0.177 | 0.263 | 0.146 | 9 |
| Autoformer | 0.177 | 0.325 | 0.157 | 0.204 | 0.376 | 0.181 | 0.143 | 0.230 | 0.122 | 0.165 | 0.263 | 0.141 | 8.5 |
| Informer | 0.248 | 0.299 | 0.211 | 0.283 | 0.324 | 0.239 | 0.119 | 0.194 | 0.102 | 0.137 | 0.217 | 0.116 | 5 |
| Reformer | 0.259 | 0.331 | 0.222 | 0.297 | 0.361 | 0.247 | 0.124 | 0.208 | 0.106 | 0.141 | 0.233 | 0.119 | 10 |
| FourierGNN | 0.097 | 0.166 | 0.086 | 0.110 | 0.186 | 0.098 | 0.084 | 0.143 | 0.073 | 0.096 | 0.164 | 0.083 | 3 |
| StemGNN | 0.155 | 0.259 | 0.137 | 0.179 | 0.285 | 0.158 | 0.128 | 0.189 | 0.109 | 0.148 | 0.206 | 0.124 | 7.5 |
| TPGNN | 0.086 | 0.152 | 0.079 | 0.099 | 0.173 | 0.090 | 0.077 | 0.139 | 0.070 | 0.089 | 0.158 | 0.081 | 2 |
| MTGNN | 0.013 | 0.031 | 0.030 | 0.016 | 0.034 | 0.038 | 0.010 | 0.020 | 0.070 | 0.012 | 0.023 | 0.093 | 2.5 |
| GraphWaveNet | 0.009 | 0.021 | 0.028 | 0.013 | 0.028 | 0.040 | 0.010 | 0.020 | 0.069 | 0.012 | 0.025 | 0.092 | 1.5 |
| GLIDE | 0.008 | 0.019 | 0.023 | 0.010 | 0.022 | 0.030 | 0.008 | 0.016 | 0.060 | 0.010 | 0.020 | 0.079 | 0.5 |

**Results on Synthetic.**   Across difficulty levels, the full model consistently yields the lowest MAE/RMSE at both short and long horizons. On the Easy split, our model is best at $H=1$ and maintains the lead through $H=12$; as difficulty increases (Medium → Hard/Very Hard), the margin becomes more visible. These trends support that (i) keeping two graphs and (ii) mixing multi-hop information are complementary; removing either part (**D1** or **D2-Random**) hurts most at short horizons, while corrupting the base topology (**D2-Random**) mainly degrades late-horizon stability.

**Results on Exchange Rate.**   We observe the same pattern. The full model (GLIDE) outperforms **D1** and **D2** for $H=3, 6, 12$. **D2-noGraph** is notably worse at $H=3$, indicating the adaptive branch is crucial for near-term predictions. **D2-Random** remains competitive at short horizons (the adaptive branch compensates for an imperfect base graph) but lags behind at $H=12$. Overall, dual-graph aggregation with power mixing is the most robust across horizons.

**Dynamic Branch Instantiations.**   We instantiate Theorem 3.1 for the dynamic branch with a box kernel of width $B$, which yields two natural choices. The *raw-signal* variant computes a short-window local similarity,

$$\hat{\mathbf{A}}_{\mathrm{d}}^{(t)} = \mathrm{ReLU}\Big( \tfrac{1}{B} \sum_{k=t-B/2}^{t+B/2} \tilde{\mathbf{x}}_k \tilde{\mathbf{x}}_k^\top \mathbf{M}_{\mathrm{d}} \Big), \tag{10}$$

a direct realisation of Theorem 3.1 with $K(s-t) = \frac{1}{B}\mathbf{1}\{|s-t| \leq B/2\}$. The *gradient* variant, defined in Equation (4), replaces $\tilde{\mathbf{x}}_k$ with the first-order temporal difference $\nabla\tilde{\mathbf{x}}_k = \tilde{\mathbf{x}}_k - \tilde{\mathbf{x}}_{k-1}$ so the dynamic branch isolates transient shocks while the static branch absorbs the slow-varying correlation. In both variants, the learned $\mathbf{M}_{\mathrm{d}}$ together with ReLU echo the non-negativity and sparsity that the Graph-LASSO penalty enforces in the theorem.

**Ablation on Dynamic Graph Construction.**   We further evaluate the effect of different dynamic graph constructions under the same training protocol. Specifically, *Data-Corr* constructs the dynamic graph from the raw-signal local correlation defined in Equation (10), while *Data-Gradient* replaces the raw signal with first-order temporal differences to emphasize transient changes defined in Equation (4). We keep model depth, optimization settings, and all other components identical across variants. For each dataset, we report MAE/RMSE at horizons $H \in 3, 6, 12$ and compute the degradation separately within each dataset. We report the result in Tab. 10.

**Conclusion.**   The results show that the preferred dynamic graph construction method depends on the dataset's temporal characteristics. On EXCHANGE RATE, the raw-signal data-correlation graph achieves slightly lower errors, especially for RMSE, suggesting that direct local correlations are sufficient when the underlying dynamics are relatively smooth and stable. In contrast, on SYN-MEDIUM and SYN-VERY HARD, the gradient-based graph consistently improves both MAE and RMSE across all prediction horizons, with average reductions of 2.1%/4.3% and 5.2%/4.4% in MAE/RMSE, respectively. This supports the motivation of using first-order temporal differences to isolate transient shocks and rapidly changing dependencies. Overall, the ablation confirms that gradient-based dynamic graphs are particularly beneficial in strongly time-varying regimes, while raw correlation graphs can remain competitive for smoother real-world signals.

*Table 8.* EXCHANGE RATE forecasting results. Prefix-mean MAE/RMSE over prediction horizons $H$. Lower is better.

| | MAE | | | RMSE | | |
|---|---|---|---|---|---|---|
| Method | H=3 | H=6 | H=12 | H=3 | H=6 | H=12 |
| D2-Random | .009 | .009 | .010 | .009 | .012 | .015 |
| D2-NoGraph | .009 | .011 | .011 | .009 | .017 | .019 |
| **D2 (Ours)** | **.005** | **.006** | **.007** | **.008** | **.009** | **.012** |

## I.1. Limitations and Future Work

Our analysis in Theorem 3.1 assumes mutually independent temporal perturbations for analytical tractability. This assumption may be violated in domains such as energy and finance, where temporal dependencies are prevalent. However, our experiments indicate that structural drift is the primary challenge under high TCV conditions, and D2 remains effective even on Germany and France (TCV > 0.96), which exhibit substantial temporal dependence. Furthermore, TCV currently relies on Pearson correlation and therefore captures only linear relationships. Future work includes extending TCV to nonlinear similarity measures, such as Dynamic Time Warping and kernel-based methods, and investigating their ability to characterize structural dynamics.

*Table 9.* Ablation study of GLIDE. Blue cells indicate the largest degradation relative to the full model (**Ours**). Removing either D1 or D2 consistently degrades performance, with D2 contributing more substantially overall

| Methods | Metrics | Exchange Rate | | | **Deg.** | Syn-Medium | | | **Deg.** | Syn-Very Hard | | | **Deg.** |
|---|---|---|---|---|---|---|---|---|---|---|---|---|---|
| | | 3 | 6 | 12 | | 3 | 6 | 12 | | 3 | 6 | 12 | |
| Data-Gradient | MAE | .006 | **.006** | **.007** | +6.7% | **.402** | **.448** | **.489** | -2.1% | **.439** | **.455** | **.465** | -5.2% |
| | RMSE | .009 | **.009** | .013 | +12.3% | **.529** | **.552** | **.562** | -4.3% | **.546** | **.547** | **.544** | -4.4% |
| Data-Corr | MAE | **.005** | **.006** | **.007** | – | .410 | .461 | .496 | – | .462 | .480 | .491 | – |
| | RMSE | **.007** | **.009** | **.012** | – | .544 | .572 | .602 | – | .564 | .571 | .578 | – |

*Table 10.* Ablation study on the construction of the dynamic graph in GLIDE. Data-Corr is slightly better on Exchange Rate, whereas Data-Gradient consistently improves performance on Syn-Medium and Syn-Very Hard, indicating its advantage in capturing rapidly changing dynamics.

| Methods | Metrics | Exchange Rate | | | **Deg.** | Syn-Medium | | | **Deg.** | Syn-Very Hard | | | **Deg.** |
|---|---|---|---|---|---|---|---|---|---|---|---|---|---|
| | | 3 | 6 | 12 | | 3 | 6 | 12 | | 3 | 6 | 12 | |
| Data-Gradient | MAE | .006 | **.006** | **.007** | +6.7% | **.402** | **.448** | **.489** | -2.1% | **.439** | **.455** | **.465** | -5.2% |
| | RMSE | .009 | **.009** | .013 | +12.3% | **.529** | **.552** | **.562** | -4.3% | **.546** | **.547** | **.544** | -4.4% |
| Data-Corr | MAE | **.005** | **.006** | **.007** | – | .410 | .461 | .496 | – | .462 | .480 | .491 | – |
| | RMSE | **.007** | **.009** | **.012** | – | .544 | .572 | .602 | – | .564 | .571 | .578 | – |

