# OpenReview forum: "When GNNs Fail: Quantifying and Overcoming Temporal Correlation Volatility in Time Series"
_ICML.cc/2026/Workshop/FMSD — FMSD @ ICML 2026 Poster_

### Official Review · Reviewer_6tge · 2026-05-22
**Review for submission 190**

**Rating:** 6
**Confidence:** 4

**Review:**

Review:

The paper studies the limitation of GNN-based multivariate time-series forecasting under dynamic inter-variable dependencies. The authors argue that many existing graph forecasting methods rely on static or globally estimated graph structures, which can fail when correlations between variables change over time. To address this, the paper introduces Temporal Correlation Volatility (TCV), a metric for quantifying topology shifts, and proposes GLIDE, a lightweight graph layer with path-based message passing and static/dynamic propagation separation. Experiments on synthetic and real-world datasets show that GLIDE consistently outperforms GNN, Transformer, and sequence-model baselines, especially on high-TCV energy datasets such as Germany and France.

Strengths:

(1) The paper addresses an important problem in multivariate time-series forecasting. Real-world dependencies between variables are often dynamic, while many existing graph forecasting models assume a fixed or globally estimated topology.

(2) The proposed TCV metric is simple and useful. It provides an interpretable way to quantify temporal changes in correlation structure and connect topology volatility with forecasting degradation.

(3) The GLIDE architecture is well motivated. Path-based message passing can reduce sensitivity to noisy direct correlations, while static/dynamic propagation separation explicitly models both persistent and transient dependencies.

(4) The empirical evaluation is strong. The paper compares against many baselines, including SpatialGNNs, SpectralGNNs, Transformers, and sequence models, and GLIDE achieves the best overall performance on both dynamic and common benchmark datasets.

Areas for Improvement:

(1) The TCV metric is intuitive, but its limitations should be discussed more carefully. Since it is based on changes in Pearson correlation matrices, it may not capture nonlinear dependencies, lagged relationships, or causal changes.

(2) The theoretical analysis is interesting, but the assumptions should be better connected to practice. For example, Gaussian perturbations and smoothly varying covariance may not fully describe abrupt event-driven topology shifts.

(3) Some empirical improvements are very large, so the paper should provide more explanation of why GLIDE improves so much over strong baselines and whether all models were tuned with comparable effort.

(4) The method would benefit from more sensitivity analysis, including the effects of local window length (B), path order (K), graph sparsification strategy, and different dynamic graph construction choices.

Detailed Comments:

(1) Please clarify what “dynamic topology” means in this work. Does it refer specifically to changing Pearson correlations, changing conditional dependence, learned adjacency variation, or broader causal structure changes?

(2) The paper should include more discussion of failure cases. For example, when does dynamic propagation add noise, and are there datasets where GLIDE does not improve over simpler graph methods?

Justification of Score:

The paper identifies an important weakness of existing graph-based forecasters, proposes a simple diagnostic metric, and introduces a well-motivated architecture for dynamic topology modeling. The experimental results are convincing and show consistent gains across several datasets. However, the theoretical assumptions need stronger practical justification, and the very large empirical gains would benefit from deeper analysis, fairness details, and sensitivity studies.